# A Systematic Review and Meta-Analysis on Metabolic Bone Disease in Patients with Primary Sclerosing Cholangitis

**DOI:** 10.3390/jcm11133807

**Published:** 2022-06-30

**Authors:** Claudiu Marinel Ionele, Adina Turcu-Stiolica, Mihaela Simona Subtirelu, Bogdan Silviu Ungureanu, George Ovidiu Cioroianu, Ion Rogoveanu

**Affiliations:** 1Doctoral School, University of Medicine and Pharmacy of Craiova, 200349 Craiova, Romania; ioneleclaudiu@gmail.com (C.M.I.); cioroianu_george@yahoo.com (G.O.C.); 2Department of Pharmacoeconomics, University of Medicine and Pharmacy of Craiova, 200349 Craiova, Romania; mihaela.subtirelu@yahoo.com; 3Research Center of Gastroenterology and Hepatology, University of Medicine and Pharmacy of Craiova, 200638 Craiova, Romania; boboungureanu@gmail.com (B.S.U.); ion.rogoveanu@umfcv.ro (I.R.); 4Department of Gastroenterology, University of Medicine and Pharmacy of Craiova, 200349 Craiova, Romania

**Keywords:** primary sclerosing cholangitis, cholestasis, osteopenia, osteoporosis

## Abstract

Data about the association between primary sclerosing cholangitis (PSC) and metabolic bone disease are still unclear. PSC is a chronic cholestatic liver disease (CCLD) which affects the biliary tract, and it has a highly variable natural history. We systematically searched until 28 February 2022 MEDLINE, Cochrane Central Register of Controlled Trials, the ISI Web of Science, and SCOPUS, for studies in patients with PSC. We identified 343 references to potential studies. After screening them, we included eight studies (893 PSC patients, 398 primary biliary cirrhosis (PBC) patients, and 673 healthy controls) for the present meta-analysis. Pooled analyses found no difference in BMD-LS (Z = 0.02, *p*-value = 0.98) between PSC patients and healthy controls. BMD-LS was statistically lower in PBC patients than in PSC patients (Mean Difference, MD, 0.06, 95% CI 0.03 to 0.09, *p*-value = 0.0007). The lumbar spine T-score was higher in the PSC patients compared with PBC patients (MD 0.23, 95% CI 0.04 to 0.42, *p*-value = 0.02). Given the limited literature available, better designed, and larger scale primary studies will be required to confirm our conclusion.

## 1. Introduction

Primary sclerosing cholangitis (PSC) is characterized by strictures in the biliary tree which lead to biliary fibrosis and progress to end-stage liver disease [1]. PSC mainly affects young and middle-aged male adults and may be associated with inflammatory bowel disease (IBD) [2,3]. The majority of PSC patients are asymptomatic for a long time. The disease may also have an aggressive course, leading to recurrent biliary tract obstruction, repetitive episodes of cholangitis, and may progress to end-stage liver disease in a shorter time [1]. Several immunologic markers and serum autoantibodies are found in most patients with PSC, although none is specific to the disease [4,5]. 

While no available therapy has proven its efficacy in stopping disease progression, the only curative option remains liver transplantation. PSC is well known as one of the leading indications in Europe for liver transplants; however, the graft loss incidence remains the highest among all other indications [6]. Moreover, it may be associated with malignancies either hepatobiliary, cholangiocarcinoma or hepatocellular carcinoma, and colorectal cancer, especially in patients with IBD [6,7,8]. 

All chronic cholestatic liver disease (CCLD) is frequently associated with osteoporosis; however, the data on PSC is scarce. There are a few risk factors associated with bone disorders including malnutrition; alcohol abuse; kidney disease; tobacco use; liver cirrhosis; neoplastic illness; prolonged glucocorticoid treatment (prednisone 5 mg/day for >3 months); some hormonal disturbances such as diabetes, hyperthyroidism, hypogonadism, and hyperparathyroidism; Cushing syndrome; hypercalciuria; and Vitamin D deficiency [9,10]. 

The prevalence of osteoporosis in patients with chronic liver disease was reported to be approximately 30% [11]. In addition, its prevalence was higher in patients with CCLD, and the prevalence of fractures was reported to be 7–35% [12]. Compared to the occurrence of osteoporosis accompanied by the low bone formation in primary biliary cholangitis (PBC), which has been reported frequently, this pattern is still unclear for PSC [13,14,15]. Osteoporosis is a health condition that weakens bones; it is a systemic skeletal disease that is characterized by low bone density and microarchitectural disruption of bone tissue, with a consequent high skeletal fragility and fracture incidence [16]. 

While it has been studied extensively in patients with PBC, bone impairment in other CCLD such as PSC may also be encountered [17]. A couple of studies have suggested that PSC might be associated with a higher risk of osteoporosis [18,19]. In the last few years, a series of developments improved our knowledge about the management of osteoporosis and its pathogenic mechanisms [20,21,22]. However, there is still no unanimity on the guidelines for the diagnosis and treatment of bone diseases as a complication of CCLD. Many controversies remain to be argued in these type of patients [23]. Therefore, additional studies should be performed on the topic of bone diseases in patients with CCLD in the future. In this paper, we highlight the definitions of related terms, guidelines for bone diseases and chronic liver disease, actual knowledge of pathogenesis, and finally target the management of bone diseases in patients with chronic liver disease. Hence, we conducted a systematic review and meta-analysis to evaluate the relationship between PSC and osteoporosis.

## 2. Materials and Methods 

We followed the 27 items for the Systematic reviews and Meta-Analysis (PRISMA) [24]. We systematically searched MEDLINE, Cochrane Central Register of Controlled Trials, the ISI Web of Science, and SCOPUS, for studies in patients with PSC. Two authors (C.M.I. and M.S.S.) independently screened until 28 February 2022, the articles published until 31 January 2022 for inclusion and extracted data. Disagreements were solved by the third author (A.T.S.). Criteria for considering studies for this meta-analysis respected the PICO standards (Patient, Intervention, Comparator, Outcomes). We included in this meta-analysis, the original articles when the following criteria were fulfilled: (1) clinical-study or cohort design; (2) PSC used as an exposure factor; (3) osteopenia, osteoporosis, or fracture as an outcome; and (4) mean differences (MD) with 95% Confidence Intervals (CI) or Odds Ratio with 95% CI were calculated for assessing the differences among the PSC patients and the controls (PBC patients or Healthy). For each group of patients, we have included only complete study analysis or the most recent largest sample size. Exclusion criteria included (1) case-report article, review article, meta-analyses, abstracts, or letters, (2) incomplete data or lack of measuring, (3) only abstract (without accessible full-text article), (4) other types of chronic liver disease than PSC or PBC, (5) studies without bone mineral density evaluation, (6) studies that involved only animals and/or ex vivo samples, (7) studies of low methodological quality, (8) studies with insufficient data, and (9) studies in languages other than English and French. Our core search consisted of the terms “primary sclerosis cholangitis” and “osteoporosis”. We extracted data on measurements for lumbar spine bone mineral density (BMD), hip BMD, T-score, Z-score, calcium, vitamin D, bilirubin, alkaline phosphatase, and fractures outcomes. When necessary, we contacted the correspondence authors to obtain additional information. Studies must report mean values of outcomes, along with SDs or SEs, or the number of events for PSC patients and controls (PBC or healthy). 

Two authors (C.M.I. and M.S.S.) independently conducted a quality assessment of included studies using the Newcastle–Ottawa Quality Assessment Scale (NOS) for observational studies (cross-sectional, case–control, or cohort), scoring them from 1 to 9 as follows: low quality (1–3), moderate (4–6), and high quality (7–9) [25,26].

Statistical analysis was performed using the Review Manager (Rev Man) software version 5.4.1 (The Cochrane Collaboration, 2020) and R package Metafor [27] (version 4.1, R foundation, Vienna, Austria). Mean differences (MD) with 95% Confidence Intervals (CI) or Odds Ratio with 95% CI were calculated for assessing the differences among the PSC patients and the controls (PBC patients or Healthy) [28]. The meta-analysis was conducted using the standardized mean difference (MD) or odds ratio (OR) as the outcome measure. We quantified Chi^2^ and its *p*-value to simply test whether effect sizes depart from homogeneity; Tau^2^ that evaluates the amount of heterogeneity; and Q-test and I^2^ that express the proportion of dispersion due to heterogeneity. In case of any level of heterogeneity, a prediction interval for the true outcome was provided. We examined if studies may be outliers and/or influential in the context of the model performing Cook’s distances. The funnel plot asymmetry is largely subjectively interpreted, so the publication bias was checked using the Begg and Mazumdar’s rank correlation test and the regression Egger’s test. A value for *p*-value less than 0.05 was considered statistically significant.

## 3. Results

### 3.1. Study Selection and Characteristics of Included Studies

The association between the risk of osteoporosis and PSC remains controversial. To solve this issue, we conducted a meta-analysis in which we identified 343 references to potential studies. After screening them as in Figure 1, we included eight studies (893 PSC patients, 398 PBC patients, and 673 healthy controls) for meta-analysis. To our knowledge, this is the first meta-analysis to explore the potential relationship between PSC and osteoporosis (Table 1).

### 3.2. Lumbar Spine BMD

#### 3.2.1. PSC vs. Healthy

BMD-LS was reported in three studies that included 300 PSC patients, as in Figure 2. The values for control healthy patients were considered from Hologic data. High heterogeneity between studies was found (Chi^2^ = 15.62, I^2^ = 87%, *p* = 0.0004). Figure 2A shows, in random effects, no difference in BMD-LS (overall effect Z = 0.27, *p* = 0.79) between PSC patients (n = 300) and healthy patients (n = 300). According to the Q-test (Q = 19.4644, *p* < 0.0001, Tau^2^ = 0.00), the outcomes were demonstrated to be heterogeneous. 

An evaluation of the studentized residuals showed that the study Wyszomirska 2015 [25] may be a potential outlier in the context of this model. None of the studies could be overly influential, according to Cook’s distances; all the studies are balanced, having almost the same weight. Egger’s regression test indicated a value of 1.149, *p* = 0.250 with no funnel plot asymmetry, as in Figure 2B. The same results were demonstrated through the Rank correlation test with value = 0.333 and *p* = 1.00.

#### 3.2.2. PSC vs. PBC

Pooling the results about the association between PSC and lumbar spine BMD, in comparison with PBC, only one study was found comparing PSC patients (n = 204) with PBC patients (n = 156). BMD of lumbar spine was statistically lower in PBC patients (MD 0.06, 95% CI 0.03 to 0.09, *p*-value = 0.0007), as in Figure 3.

### 3.3. Lumbar Spine T-Score

As shown in Figure 4, the lumbar spine T-score was higher in the 436 PSC patients than the 383 PBC patients (MD = 0.23, 95%CI 0.04 to 0.42). The data from the two studies were pooled using a fixed effects model since no heterogeneity was found (I^2^ = 0%, Chi^2^ = 0.37, *p* = 0.54).

### 3.4. Lumbar Spine Z-Score

Two studies examined the values of the Z-score for PSC patients (n = 436) and PBC patients (n = 383). We pooled the data to evaluate the association between PBC and Z-score with a fixed-effects model analysis, as in Figure 5. There was a trend to a smaller value of z score for PSC patients than for PBC patients (SMD −0.15, 95%CI −0.34 to 0.03, *p* = 0.11).

### 3.5. Hip BMD

Concerning hip BMD, no studies were found comparing PSC patients with PBC patients, but only PSC patients with healthy patients as control group. Angulo et al. demonstrated statistically higher hip BMD for PSC patients than for healthy people, as in Figure 6.

#### 3.5.1. Vitamin D PSC vs. Healthy

A number of five studies was included in the meta-analysis with 610 PSC patients and 610 healthy patients. The amount of heterogeneity is high (Tau^2^ = 3.17, Q = 12.608, I^2^ = 69%, *p* = 0.01) and a random-effects model was fitted to the data, as in Figure 7A. Vitamin D level did not differ significantly between PSC patients and healthy patients. Although the average outcome is estimated to be positive (MD = 1.00, 95%CI −0.99 to 3.00), in some studies, the true outcome may in fact be negative. An evaluation of the studentized residuals revealed that the study Hay 1991 may be a potential outlier in the context of this model. None of the studies could be considered to be overly influential according to Cook’s distances; Angulo et al. had the biggest weight.

Egger’s regression value (−1.343) and its *p*-value (0.179) together with its Rank correlation test value (−0.600) and its *p*-value (0.233) did not indicate any funnel plot, as in Figure 7B.

#### 3.5.2. Calcium PSC vs. Healthy

A total of three studies were included in this analysis. There was a trend towards lower calcium level for PSC patients than for healthy ones (SMD −0.29, 95% CI −0.59 to 0.00, *p* = 0.05), with the majority of estimates being negative. According to the Q-test, the studies appear to be heterogeneous (Tau^2^ = 0.07, Chi^2^ = 55.12, Q = 41.0255, *p* < 0.00001, I^2^ = 96%). A 95% prediction interval for the true outcomes is given by −3.4148 to 1.4321. Hence, although the average outcome is estimated to be negative, in some studies the true outcome may in fact be positive. According to the studentized residuals, the study Keller et al. [26] may be a potential outlier in the context of this model, as in Figure 8A. An examination of Cook’s distances revealed none of the studies could be considered to be overly influential, the weight of them being balanced.

As in Figure 8B, neither the rank correlation (−0.333, *p* = 1.00) nor the Egger’s regression test (−1.687, *p* = 0.0915) indicated any funnel plot asymmetry. 

#### 3.5.3. Bilirubin PSC vs. Healthy

A total of six studies were included in the analysis. According to the Q-test, the outcomes appeared heterogeneous (Tau^2^ = 6.37, Q = 698.64, *p* < 0.00001, I^2^ = 99%) and the random-effects model was estimated, as in Figure 9A. The bilirubin was found to be higher in PSC patients. An examination of the studentized residuals revealed there was no indication of outliers in the context of this model. None of the studies could be overly influential. According to Cook’s distances, all the studies have almost the same weight, 16–17%.

Both Rank correlation test (value = 0.867, *p* = 0.017) and Egger’s regression test (value = 2.134, *p* = 0.033) indicated potential funnel plot asymmetry, as in Figure 9B.

#### 3.5.4. Alkaline Phosphatase PSC vs. Healthy

A total of five studies were included in this analysis, as in Figure 10A. The pooled data revealed a significantly higher value of alkaline phosphatase for PSC patients than for healthy ones. According to the Q-test (Tau^2^ = 292798.79, Q = 796.386, Chi^2^ = 1580.88, *p* < 0.00001, I^2^ = 100%), the studies are heterogenous and the random-effects model was used. An examination of the studentized residuals revealed that Angulo et al. may be a potential outlier in the context of this model. 

Checking for funnel plot asymmetry, the Egger’s regression test indicated asymmetry (7.969, *p* < 0.001), but not the Rank correlation test (0.40, *p* = 0.483). The funnel plot from Figure 10B suggests the presence of publication bias as smaller studies (which appear toward the bottom) were not included into this analysis.

### 3.6. Prevalence of Fractures

Two studies inspected the effects of PSC compared with PBC on fractures, including 442 PSC patients and 398 PBC patients. We observed low heterogeneity between the studies (Chi^2^ = 0.01, *p*-value = 0.92, I^2^ = 0%). Pooling the results with a fixed-effects model, as in Figure 11, the tendency was to report less fracture events in PSC patients than in PBC patients (OR = 0.68, 95% CI 0.46 to 1.00, *p*-value = 0.05). 

## 4. Discussion

CCLD increases the risk of mineral bone depletion, particularly in PBC and PSC [35]. In recent years, a number of studies have indicated that osteoporosis occurs at a substantially higher rate among PSC patients than in age- and sex-matched healthy controls [31]. The incidence of bone disease was associated with elevated Mayo risk scores and advanced histological stages [14]. It remains uncertain when and how often the bone density measurement needs to be repeated in patients with PSC, but it is necessary to screen these patients at diagnosis and every 2 to 3 years for bone mineral deficiency.

Our meta-analysis does not show a potential link between PSC and osteoporosis, but the heterogeneity of our studies (different age for the patients included in our studies: mean age ± SD, 35.3 ± 13.38 [34] vs. 45.5 ± 0.8 [19], 46.9 ± 13.4 [18]) could be the reason for our outcomes.

We found in particular, a higher prevalence of osteoporosis in PSC patients when compared with PBC, but not with a significantly reduced Z-score (*p*-value = 0.11). The prevalence of osteoporosis in CCLD patients ranges from as low as 13% to as high as 95% [3,36]. A higher prevalence is reported in the Indian population, between 68 and 95% versus 13 and 70% in Western countries [12]. Bone turnover laboratory parameters showed noticeably increased levels of ALP and high bilirubin levels in PSC patients. We reviewed data from five clinical studies and three cohort studies with PSC vs. PBC patients and PSC vs. healthy controls, including 398 PBC patients and 893 PSC patients. Although most included studies reported a positive trend, each study had a relatively small sample size.

Along with liver disease advancement, lower levels of vitamin D might be encountered. This was studied by Hay et al. who highlighted that advanced PSC patients have a lower level of vitamin D than newly diagnosed PSC patients. However, they did not correlate the presence of osteopenia [37]. Moreover, when an osteoporosis stage is reached, vitamin D levels will be even lower [19]. In the current meta-analysis, we assessed vitamin D and calcium levels and found a lower trend for calcium levels in patients with PSC than for healthy patients, with no significant differences in vitamin D in line with the current studies. This is likely due to the small homogeneity for the patients included in the studies.

A noteworthy finding of our study is that the T-score is significantly increased in PSC adults than in PBC adults, whereas our pooled data demonstrated no differences in Z-scores. These results are similar to other studies which quantified the T-score in the lumbar spine suggesting discrepancies between PBC and PSC, with significantly lower values. In addition, Janes et al. pointed out that in PBC patients, bone resorption was lower than in patients diagnosed with PSC [38]. 

In recent years, the term “hepatic osteodystrophy” [39,40] has been used to define the presence of bone disorders in cirrhotic patients with cholestasis. In this context, we found that bilirubin values were significantly higher in patients with PSC compared with controls. It is accepted that cholestasis can contribute to bone impairment in patients with CCLD leading to metabolic bone abnormalities and osteoporosis. Serum bilirubin did correlate inversely with serum osteocalcin [29,41]. The incidence of osteodystrophy seems to be directly related to the body mass index decrease, with increasing disease duration, age, as well as disease severity, although the last factor was not confirmed and fully understood [33,40,41]. 

Only one study [30] reported higher BMD–LS in PSC than PBC, which might be related to the severity index for liver disease as assessed by a score based on multiple risk factors exemplified in 210 PBC [42] and 37 PSC patients [43]. However, controversial results were pooled in our meta-analysis from studies comparing PSC patients with healthy adults, indicating no differences of BMD-LS between them.

We found significantly higher alkaline phosphatase values related to PSC patients. de Vries et al. [44] demonstrated that ALP predicts the prognosis in PSC. The data indicated that values at 1-year post-diagnosis have slightly higher predictive power than at diagnosis, justifying further studies to explore the use of ALP as a surrogate endpoint for clinical trials.

Our meta-analysis showed a lower risk of bone fractures associated with PSC than with PBC. However, these findings remain uncertain due to the small number of studies and should be addressed in future studies. However, we suggest the need for more specific prevention and treatment strategies for osteoporosis. If osteopenia is detected in patients with a high risk of bone disease, they should be treated with daily vitamin D 400 IU (10 μg) and calcium supplements if calculated dietary calcium intake is insufficient [45]. This collected data can be used for feature guidelines regarding the frequency of bone demineralization screening in these patients. More than half of patients with PSC present with a T-score in the range of osteopenia or osteoporosis. Therefore, it seems reasonable to measure the bone density in all patients at the time of diagnosis of PSC.

The main limitations were the small number of included studies and their heterogeneity. The small number of studies did not permit performing a meta-regression to test differences between subgroups. In addition, we did not perform subgroup analysis according to duration and severity of cholestasis, histological stage, or bone turnover laboratory parameter analysis, due to insufficient information and limited studies. This can be attributed to the heterogeneity of the studies, which included patients with inflammatory bowel disease, menopausal status for females, smoking history, current alcohol consumption, and hepatic decompensation (ascites, variceal bleeding, or hepatic encephalopathy). The included studies evaluated advanced and newly PSC patients without differentiating the outcomes for each group. In the present article, we discovered that only five out of eight articles briefly mentioned the use of steroids for IBD patients. We saw in Table 1 that the studies included in this meta-analysis mentioned use of steroids, cumulative dose or a mg per day dose. We presume that all the indications for corticosteroid administration were in line with the guidelines for IBD patients [46]. Medication exposure was recorded including intake of bisphosphonates, vitamin D supplements, oral administration of steroids, ursodeoxycholic acid, calcium supplements, prednisone, oral contraceptives, and hormone replacement therapy [2,6,18,19,30,33].

## 5. Conclusions

In conclusion, the current meta-analysis demonstrates that PSC is not significantly associated with osteoporosis compared to healthy controls. Patients of a younger age do not impact BMD. However, we observed different results of BMD-LS between PSC and PBC, with lower rates of fractures for PSC patients. In order to benefit from an accurate screening of bone status and for more specific prevention strategies, more clinical management should be performed with caution in PSC patients. Therefore, based on the limited number of studies included, additional and specifically designed studies are needed in order to extend our results and determine the mechanisms underpinning the relationship between PBC and osteoporosis risk.

## Figures and Tables

**Figure 1 jcm-11-03807-f001:**
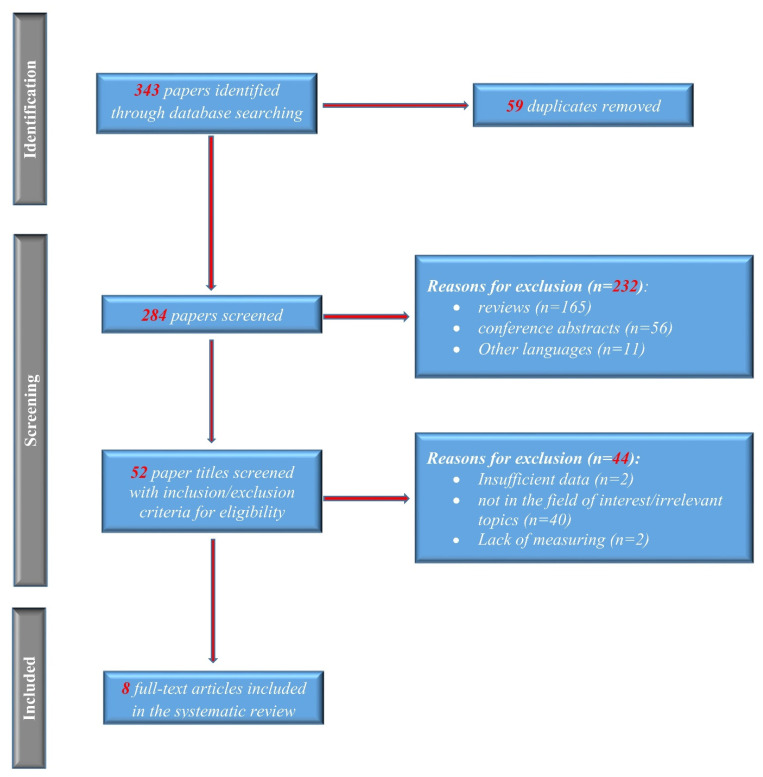
PRISMA flow of the selection process.

**Figure 2 jcm-11-03807-f002:**
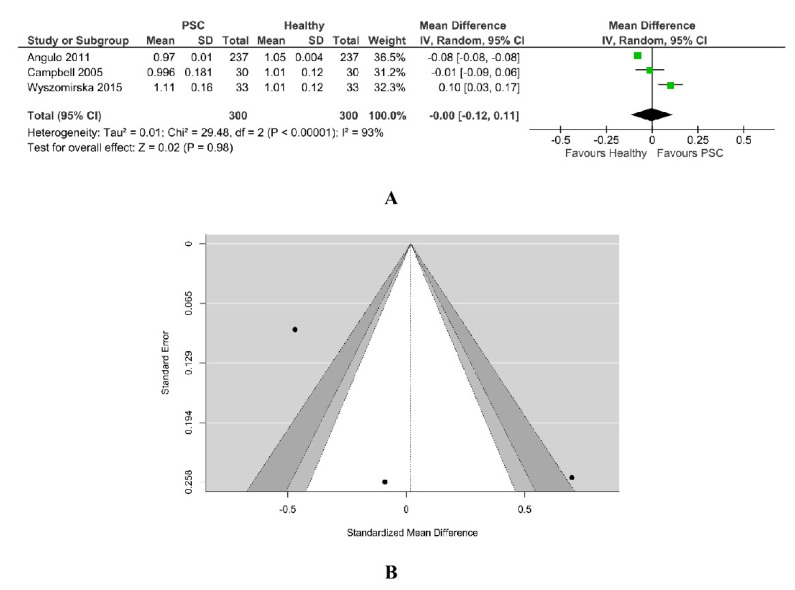
(**A**) Forest plot assuming a random-effects model for BMD-LS PSC vs. Healthy. (**B**) Funnel plot demonstrating publication bias assessment in model for BMD-LS PSC vs. Healthy [18,19,34].

**Figure 3 jcm-11-03807-f003:**
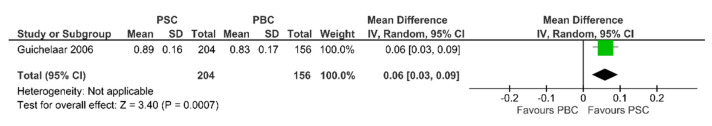
Forest plot assuming a random-effects model for BMD-LS PSC vs. PBC [30].

**Figure 4 jcm-11-03807-f004:**
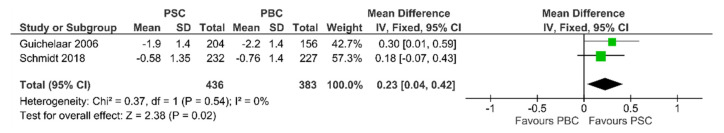
Forest plot assuming a fixed-effects model for T-score PSC vs. PBC [30,33].

**Figure 5 jcm-11-03807-f005:**
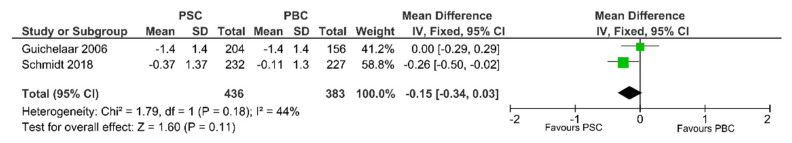
Forest plot assuming a fixed-effects model for Z-score PSC vs. PBC [30,33].

**Figure 6 jcm-11-03807-f006:**
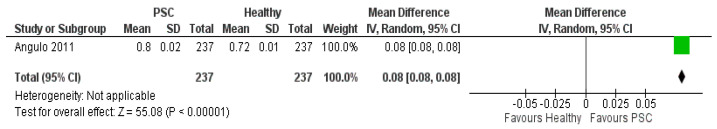
Forest plot assuming a random-effects model for BMD-hip PSC vs. Healthy [19].

**Figure 7 jcm-11-03807-f007:**
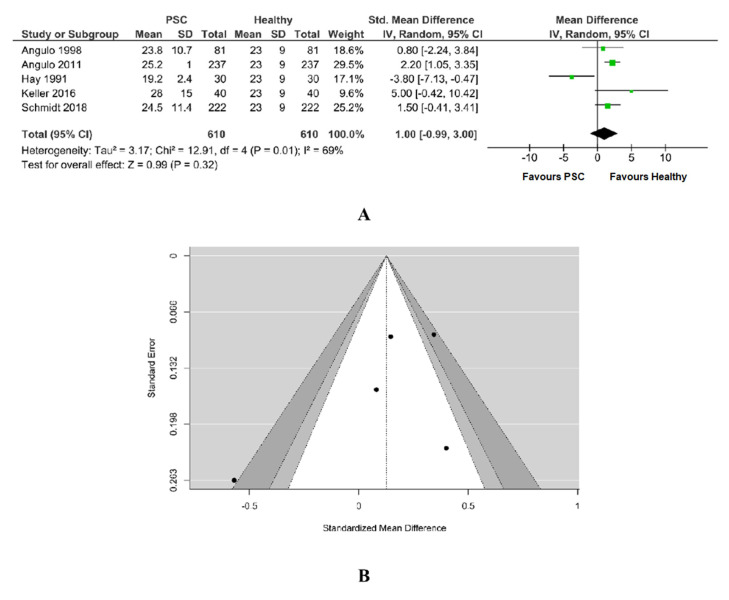
(**A**) Forest plot assuming a random-effects model for Vitamin D PSC vs. Healthy. (**B**) Funnel plot for publication bias assessment for vitamin D in PSC patients vs. Healthy [19,29,31,32,33].

**Figure 8 jcm-11-03807-f008:**
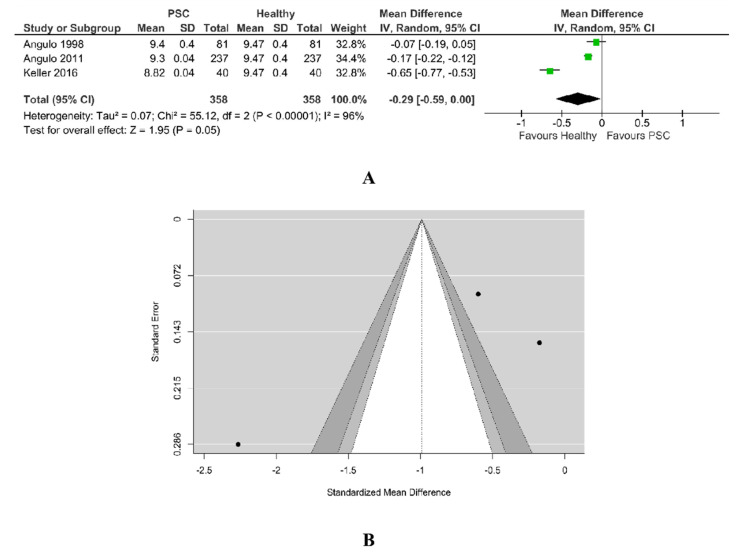
(**A**) Forest plot assuming a random-effects model for calcium PSC vs. Healthy. (**B**) Funnel plot for publication bias assessment for calcium in PSC patients vs. Healthy [19,29,32].

**Figure 9 jcm-11-03807-f009:**
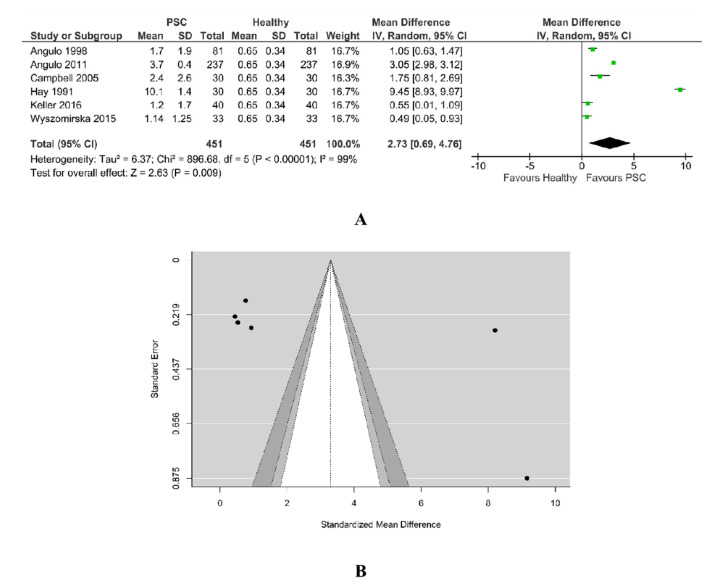
(**A**) Forest plot assuming a random-effects model for bilirubin PSC vs. Healthy. (**B**) Funnel plot for publication bias assessment for bilirubin in PSC patients vs. Healthy [18,19,29,31,32,34].

**Figure 10 jcm-11-03807-f010:**
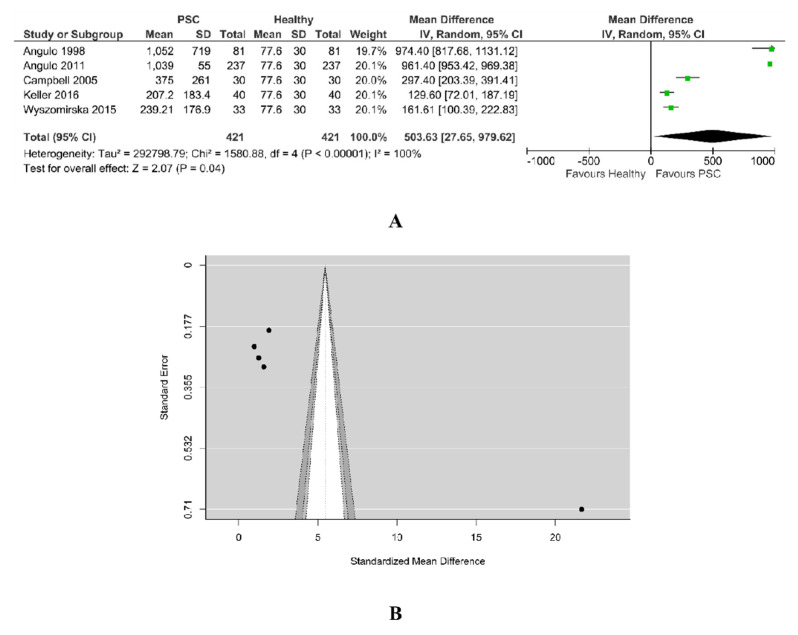
(**A**) Forest plot assuming a random-effects model for alkaline phosphatase PSC vs. Healthy. (**B**) Funnel plot for publication bias assessment for alkaline phosphatase in PSC patients vs. Healthy [18,19,29,32,34].

**Figure 11 jcm-11-03807-f011:**
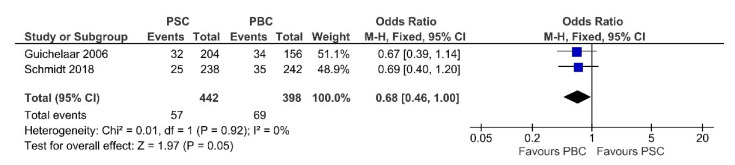
Forest plot assuming a fixed-effects model for fractures PSC vs. PBC [30,33].

**Table 1 jcm-11-03807-t001:** Description of the characteristics of included studies.

Study (Year)	Country	Study Type	Gender Male/Female PSC, Control	Age (Mean), Years PSC, Control	Mean Duration of Disease (Months)	Severity of Disease		Treatment	Quality Assessments (NOS)	Number of PSC Patients/Controls	Outcomes	Osteodensitometry Machine Used
						PSC Mayo Risk Score	MELD PSC, Control					
Angulo P et al., 1998 [29]	USA	Clinical study	44/37/NA	42.9 ± 11.5	31 ± 34, NA	2.9 ± 1.2	NA	NA	8	81/81	Prevalence of osteoporosis, lumbar spine BMD	Hologic
Angulo P et al., 2011 [19]	USA	Cohort study	42%/58%, NA	45.5 ± 0.8, 45.5 ± 0.8	63.4 ± 4.6, NA	1.19 ± 0.09	NA	Budesonide, biphosphonates (Cumulative prednisone dose)	9	237/237	Prevalence of fractures, osteoporosis, lumbar spine BMD and T-score, hip BMD and T-score	Hologic
Campbell MS et al., 2005 [18]	USA	Clinical study	25/5, NA	46.9 ± 13.4, 46.9 ±13.4	NA	1.52 ± 1.07	10.5 ± 4.5	Biphosphonates, steroids (no doses mentioned), calcium, vitamin D, ursodeoxycholic acid	8	30/30	Lumbar spine BMD	Lunar and Norland
Guichelaar MMJ et al., 2006 [30]	USA	Cohort study	142/ 218	46.8 ± 11.0/53.2 ± 8.6	85.2 ± 63.6 /94.8 ± 63.6	NA	17.0 ± 8.7/17.6 ± 8.8	Glucocorticoids (prednisone 10 mg per day), biphosphonates, ursodeoxycholic acid	9	204/156	Prevalence of fractures and osteoporosis, lumbar spine BMD and T-score	Hologic
Hay JE et al., 1991 [31]	USA	Clinical study	19/11, 13/5	39.4 ± 0.3, 42.7 ± 0.7	91.44, NA	NA	NA	NA	8	30/18	Prevalence of osteopenia, lumbar spine BMD	-
Keller S et al., 2016 [32]	Germany	Clinical study	20/20, NA	50 ± 12.6, 49.5 ± 13.0	102 ± 98.4, NA	NA	NA	Ursodeoxycholic acid, prednisolone (10 mg per day)	8	40/10	Prevalence of osteoporosis, lumbar spine BMD and T- score	GE Lunar
Schmidt T et al., 2019 [33]	Germany	Cohort study	136/ 102, 132/ 210	47.1 ±13.8	79.1 ± 75.4/71.9 ± 62.7	NA	NA	Glucocorticoids, biphosphonates, ursodeoxycholic acid, Prednisolone (no doses mentioned)	8	238/242	Prevalence of fractures, osteoporosis, lumbar spine BMD and the femoral neck	GE Lunar
Wyszomirska JR et al., 2015 [34]	Poland	Clinical study	22/11, NA	35.3 ± 13.38	40.56 ± 24.2	NA	NA	NA	8	33/33	Prevalence of fractures	Hologic

## Data Availability

Not applicable.

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
