# Peer review of "A Systematic Review and Meta-Analysis on Metabolic Bone Disease in Patients with Primary Sclerosing Cholangitis"

_jcm, 2022, doi:10.3390/jcm11133807_

Round 1

Reviewer 1 Report

Ionele et al. focused on the summary of relationship between PSC and bone disease. They performed the meta-analysis of identified 343 papers, including finally 5 studies to the overall assessment (610 PSC patients and 610 healthy patients). The Authors didn't find finally any significant differences in the explored topic. The data on this issue turned out to be quite limited.

But I'd like Authors to write more about the prevalence of bone pathologies in the course of chronic liver disorders in the introduction of manuscript.

Author Response

We are very grateful for the constructive comments from you. We also thank you for your time and effort in reviewing our manuscript.

We have carefully addressed point-by-point all the comments and made corrections in our manuscript using tracked changes, new references have been inserted in the text of the paper and the introduction was updated with information about the prevalence of bone pathologies in chronic liver disorders as follows:

We added more information about the prevalence and risk factors for bone disorders in the introduction of the manuscript (lines 48-56)

             There are a couple of risk factors associated with bone disorders including malnutrition, alcohol abuse, kidney disease, tobacco use, liver cirrhosis, neoplastic illness, prolonged glucocorticoid treatment (prednisone 5 mg/day for >3 months), some hormonal disturbances such as diabetes, hyperthyroidism, hypogonadism, hyperparathyroidism, Cushing syndrome, hypercalciuria and vitamin D deficiency [9, 10]

              The prevalence of osteoporosis in patients with chronic liver disease was reported to be approximately 30% [11]. In addition, its prevalence was higher in patients with CCLD, and the prevalence of fractures was reported to be 7–35% [12].

  1. Lopez-Larramona, G.; Lucendo, A. J.; Gonzalez-Castillo, S.; Tenias, J. M. Hepatic osteodystrophy: an important matter for consideration in chronic liver disease, W. J. Hep. 2011 3, 12, 300–307. doi:10.4254/wjh.v3.i12.300
  2. Lupoli, R.; Di Minno, A.; Spadarella, G.; Ambrosino, P.; Panico, A.; Tarantino, L.; Lupoli, G.; Lupoli, G.; Di Minno, M. N. D. The risk of osteoporosis in patients with liver cirrhosis: a meta-analysis of literature studies, Clin. Endocrinol., 2016, 84, 1, 30– 38. doi:10.1111/cen.12780
  3. Karoli, Y.; Karoli, R.; Fatima, J.; Manhar, M. Study of Hepatic Osteodystrophy in Patients with Chronic Liver Disease, J. Clin. Diag. Res. 2016, 10, 08; doi:10.7860/JCDR/2016/21539.8367
  4. Ranjan, R.; Rampal, S.; Jaiman, A.; Tokgöz, M. A.; Koong, J. K.; Ramayah, K.; Rajaram, R.; Common musculoskeletal disorders in chronic liver disease patients, J. Dis. Rel. Surg. 2021, 32, 3, 818-823, doi: 10.52312/jdrs.2021.25

Reviewer 2 Report

This review article has described the relationship between metabolic bone disease and PSC. The paper is well written according to the proper method. However, there are some concerns about this article.

1. The references are mainly derived from Western studies. How about the studied form oriental? In addition, racial differences should be considered. They should discuss these.

2. There could be a significant bias in steroid use by IBD patients for osteoporosis. The authors should discuss more precisely the detail of the ratio of steroid use from selected articles, not just in limitations.

3. The references are few and not updated for the full review article. 

4, Each figure is separated and hard to read. The authors could unite some of these.

Author Response

Many thanks for the interesting and stimulating comments and suggestions, also the authors would like to thank you for the time to observe the errors and inform us to clarify/correct the paper. As a general comment, in order to improve the clarity of the paper, changes were made to the content of the paper.

This review article has described the relationship between metabolic bone disease and PSC. The paper is well written according to the proper method. However, there are some concerns about this article.

  1. The references are mainly derived from Western studies. How about the studied form oriental? In addition, racial differences should be considered. They should discuss these.

Thank you for your recommendation. Our study aimed to assess the prevalence of bone disorders as osteopenia, osteoporosis (lines 329-330) or bone fractures (lines 391-393) between PSC vs. Healthy controls or PSC vs. PBC patients in the Discussion section. We are grateful for it and therefore, we added in the Discussion section (lines 330-333) some information about data obtained in eastern part of the world:

The prevalence of bone disease in CCLD patients ranges from as low as 13% to as high as 95% [33,34]. A higher prevalence is reported in the Indian population, between 68 to 95% versus 13 to 70% in Western countries [12].

  1. Rouillard S, Lane NE. Hepatic osteodystrophy. Hepatology 2001; 33, 1, 301-7. doi: 10.1053/jhep.2001.20533.
  2. George, J.; Ganesh, H. K.; Acharya, S.; Bandgar, T. R.; Shivane, V.; Karvat, A.; Bhatia, S. J.; Shah, S.; Menon, P.S.; Shah, N. Bone mineral density and disorders of mineral metabolism in chronic liver disease. World. J. Gastroenterol. 2009; 15: 3516-22, doi:10.3748/wjg.15.3516

  1. There could be a significant bias in steroid use by IBD patients for osteoporosis. The authors should discuss more precisely the detail of the ratio of steroid use from selected articles, not just in limitations.

              Thank you very much for this important comment. The following text has been inserted to complete the information about steroid treatment in PSC patients diagnosed also with IBD (lines 410-414): “In the present article we discovered that only 5 out of 8 articles briefly mentioned the use of steroids for IBD patients. We saw in table 1 that the studies included in this meta-analysis mentioned use of steroids, cumulative dose or a mg per day dose. We presume that all the indications for corticosteroid administration were in line with the guidelines for IBD patients [7].

  1. Lamb, C.A.; Kennedy, N.A.; Raine, T.; Hendy, P. A.; Smith, P. J.; Limdi, J. K.; Hayee, B.; Lomer, M. C. E.; Parkes, G. C.; Selinger, C. et al. British Society of Gastroenterology consensus guidelines on the management of inflammatory bowel disease in adults. Gut 2019; 68, 1–106. doi:10.1136/GUTJNL-2019-318484

3.The references are few and not updated for the full review article.

              We are very grateful for your words and we added more references from more recent studies concerning metabolic bone disease in patients with primary sclerosing cholangitis

  1. Lupoli, R.; Di Minno, A.; Spadarella, G.; Ambrosino, P.; Panico, A.; Tarantino, L.; Lupoli, G.; Lupoli, G.; Di Minno, M. N. D. The risk of osteoporosis in patients with liver cirrhosis: a meta-analysis of literature studies, Clin. Endocrinol., 2016, 84, 1, 30– 38. doi:10.1111/cen.12780
  2. Karoli, Y.; Karoli, R.; Fatima, J.; Manhar, M. Study of Hepatic Osteodystrophy in Patients with Chronic Liver Disease, J. Clin. Diag. Res. 2016, 10, 08; doi:10.7860/JCDR/2016/21539.8367
  3. Ranjan, R.; Rampal, S.; Jaiman, A.; Tokgöz, M. A.; Koong, J. K.; Ramayah, K.; Rajaram, R.; Common musculoskeletal disorders in chronic liver disease patients, J. Dis. Rel. Surg. 2021, 32, 3, 818-823, doi: 10.52312/jdrs.2021.25

4.Each figure is separated and hard to read. The authors could unite some of these.

              Thank you for this important suggestion and as we discussed vitamin D and calcium levels in lines 347-355, we agree with the figures corresponding to their forest plots to be closer so they could be easier to read.